# Early Neuro-Psychomotor Therapy Intervention for Theory of Mind and Emotion Recognition in Neurodevelopmental Disorders: A Pilot Study

**DOI:** 10.3390/children9081142

**Published:** 2022-07-29

**Authors:** Elisa Giangiacomo, Maria Castellano Visaggi, Franca Aceti, Nicoletta Giacchetti, Melania Martucci, Federica Giovannone, Donatella Valente, Giovanni Galeoto, Marco Tofani, Carla Sogos

**Affiliations:** Department of Human Neurosciences, Sapienza University of Roma, 00185 Rome, Italy; maria.castellanovisaggi@uniroma1.it (M.C.V.); franca.aceti@uniroma1.it (F.A.); nicoletta.giacchetti@uniroma1.it (N.G.); melania.martucci@uniroma1.it (M.M.); federica.giovannone@uniroma1.it (F.G.); donatella.valente@uniroma1.it (D.V.); giovanni.galeoto@uniroma1.it (G.G.); marco.tofani@uniroma1.it (M.T.); carla.sogos@uniroma1.it (C.S.)

**Keywords:** neurodevelopmental disorder, disability, theory of mind, emotions, early rehabilitation, neuro-psychomotor therapy, childhood

## Abstract

The aim of the present study is to explore the effect of early neuro-psychomotor therapy to improve theory of mind skills and emotion recognition in children with neurodevelopmental disorders. A pilot study was set up, consisting of in-group training activities based on the neuro-psychomotor approach. Children were evaluated using Neuropsychological Assessment for Child (Nepsy-II), Test of Emotion Comprehension (TEC), and Social Communication Questionnaire (SCQ). For data analysis, one-sample Wilcoxon signed rank test was used with a significance of *p* < 0.05. Two children with a developmental language disorder and four children with autism spectrum disorders participated in a 3-month training program. Our findings revealed significant improvement in emotion recognition, as measured with Nepsy-II (*p* = 0.04), while no statistical improvement was found for theory of mind. Despite the limited sample, early neuro-psychomotor therapy improves emotion recognition skills in children with neurodevelopmental disorders. However, considering the explorative nature of the study, findings should be interpreted with caution.

## 1. Introduction

Theory of mind (ToM) refers to the ability to know and attribute mental states (perceptions, emotions, and thoughts) in oneself and others. It ranges, with a development across time, from understanding simple desires and emotions to more complex mental states [1,2]. As data suggest, children at 2 years of age already manifest reasoning about desire, before the ability to reason about cognition, referring in particular to false beliefs [3]. Moreover, ToM is strongly linked to the concept of empathy, which can be divided into two domains: the social cognitive understanding of other people’s mental states (i.e., theory of mind) and empathic responsiveness, better explained as behavioral aspects of the individuals [4,5,6,7]. Evidence demonstrates that delays of development of ToM skills have been seen in deaf and hard of hearing children born to hearing parents, suggesting that environmental factors, such as language and verbal interaction, can influence its development [8]. Observational studies show that socioeconomic status [9], family size, and number of siblings [10] can have an impact on ToM. A meta-analysis of controlled studies analyzed the effectiveness of ToM training among children with an average age of 63 months [11], highlighting how early training has a positive effect on ToM skills in children. The length of the training period significantly influenced the overall effect size, while the number of sessions and gender moderate it only marginally. Common interventions among the different studies were based on training procedures, including corrective feedback, use of imagination, modeling, and role-play, all of which are similar to ToM tasks. In some studies, ToM training was considered as a part of a broader project aiming to develop social skills. Several studies, regarding individual differences of ToM acquisition, demonstrated that among typically developing children, the use of language is an important instrument in order to draw their attention to mental states [12,13,14]. Children’s language ability, from the perspective that it is needed for everyday conversation [15], and their involvement in pretend play (in which they need to assume the other’s perspective) [16] have been associated with ToM abilities in previous studies.

Facial recognition impacts social functioning, peer relationships, and behavior, and is the ability to determine facial emotions or differentiate familiar and unfamiliar faces [17]. This is important for successful interaction with others and to actively participate in the social environment and is also vital for the development of the perceptual components of ToM [18]. There are six universally recognized basic expressions of facial affect; happiness, sadness, anger, fear, surprise, and disgust [19]. The standardized facial emotional expression set [20] developed by Ekman and coworkers has identified the discrete emotions of happiness, surprise, anger, fear, disgust, and sadness. Contrasting discrete emotional states (happiness vs. sadness, amusement vs. disgust, calmness vs. excitement, calmness vs. anger, and fear vs. anger) revealed high association between emotional arousal and neuro-functional brain connectivity measures [21]. Neural networks underlying this ability involve a set of structures that includes the visual cortex, the orbitofrontal cortex, the insula, and the basal ganglia, but mainly the mesial temporal structures, with an important role played by the amygdala [22]. The recognition of others’ emotions and mental states relies on the integration of emotional cues from various channels, including facial expressions, vocal intonation, and body language, as well as contextual information [23]. However, there are differences in the recognition of basic and complex emotions between children with neurodevelopmental disorders and typically developing children [24,25]. Complex mechanisms underlying emotion recognition and ToM should be addressed by specific rehabilitation interventions.

Neuro-psychomotor therapy is a unique rehabilitation approach in Italy and worldwide. It has been referred to as psychomotricity in other European countries and it is also known as play therapy [26]. In the United States, a similar treatment may be represented in developmental therapy [27]. This approach focuses on the development of how children process information through movement during play sessions. Specifically, it proposes the following objectives: to favor the appearance of social signalers (eye contact, reference looks, smiles, etc.), to increase attention time, to facilitate a more appropriate use of objects, to stimulate communication, to enrich vocabulary, and to discourage certain behaviors (hyperactivity, motor stereotypes, self-injurious behaviors, etc.). Furthermore, psychomotricity may be considered a safe and effective therapy for neurodevelopmental disorders [28,29]. 

The objective of the present pilot study is to evaluate the efficacy of early neuro-psychomotor therapy intervention for the development of ToM skills and emotion recognition in children with neurodevelopmental disabilities.

## 2. Materials and Methods

The present study was conducted within the Child Neuropsychiatry Unit of the Department of Human Neurosciences of Sapienza University of Rome. This work was carried out on the basis of previous studies conducted by the authors who have great experience in child neuropsychiatry diagnoses and rehabilitation [30,31,32,33,34,35,36,37,38].

### 2.1. Sampling

A convenience sample was selected from the Child Neuropsychiatry Unit of the Department of Human Neurosciences of Sapienza University of Rome. Children with developmental language disorder (DLD) and autism spectrum disorder (ASD) were enrolled in the study. Compared with the general language profile of DLD, where difficulties are traditionally thought to center on structural aspects of language, in ASD, language difficulties are wider, affecting areas of pragmatics and social interaction [39]. Although the diagnostic criteria make a clear distinction between DLD and ASD (they exclude each other), co-occurrence and overlap of ASD and DLD in social, communication, and language development are discussed in many studies [40,41,42,43]. To be included in the study, children had to have the following characteristics: aged between 42 and 72 months and a diagnosis of DLD and ASD. Children with cognitive impairment (IQ < 70) and/or sensory disorders were excluded. To evaluate cognitive abilities and global functioning, different tools were used: Griffiths III [44] (Griffiths Mental Development Scales—Extended Revised [45]) and Wechsler Preschool and Primary Scale of Intelligence (WPPSI) III [46].

### 2.2. Procedures, Measures, and Data Analysis

Children were tested individually by clinicians for approximately 30–45 min in two different sessions. For measuring theory of mind tasks, emotional behavior, and social communication, the following instruments were used:

*Neuropsychological Assessment for Child.* (Nepsy-II) [47] is a comprehensive instrument designed to assess neuropsychological development in preschool and school-age children. This tool is composed of three different sessions: S01, made of 15 short stories to analyze first- and second-order beliefs, S01.1, with 8 questions in which the child must guess the character’s facial expression based on the situations depicted in the image, and S02 (16 questions for children aged 3–4, 35 for older children), in which the aim is to couple similar emotions among the pictures represented. This test can be used in children from 3 to 16 years old. NEPSY-II asks children to recognize different facial expressions and to interpret other’s beliefs and intentions and how these can affect their behaviors.

*Test of Emotion Comprehension* (TEC) [48] is an emotional-perspective taking test. It assesses the understanding of emotions in children aged 3–11 years and includes nine components: recognition, external cause, desire, belief, reminder, regulation, hiding, mixed, and morality. It consists of an A4 picture book (versions for boys and girls) presenting a series of cartoon scenarios at the top of each page; the bottom of the same page shows four possible emotional outcomes depicted by facial expressions. The child is read a short story while looking at the cartoon scenario, then the child is asked to point to the appropriate facial expression (the answer is typically nonverbal). The total score is determined by assigning 1 point for each component answered correctly.

*Social Communication Questionnaire (SCQ)* [49]. The SCQ is a brief, 40-item, parent-report screening measure that focuses on items relating to ASD symptomatology likely to be observed by a primary caregiver. Although the SCQ is a screening tool—and, thus, cannot be used for diagnosis of ASD—it is based on the Autism Diagnostic Interview (ADIR) [17], a semi-structured parent interview conducted by a trained clinician or researcher that can be used for diagnostic evaluation of children with suspected ASD. Each item in the SCQ requires a dichotomous “yes”/“no” response, and each scored item receives a value of 1 point for abnormal behavior and 0 points for absence of abnormal behavior/normal behavior.

All these measures were used for evaluation. Children repeated the same tests at the end of the intervention, three months later, while parents were given a similar set of questions from the SCQ [31], focused only on the last three months. For data analysis, one-sample Wilcoxon signed rank test was used with a significance of *p* < 0.05.

### 2.3. Neuro-Psychomotor Therapy Protocol

This study involved residents MD in child neurology, supervisors that specialized in Child Neurology and Psychiatry and neuro-psychomotor therapists. The sessions consisted of two hours twice a week, divided as follows: 30 min intervention focused on ToM or emotion understanding, and 1.30 h intervention focused on social interaction, language skills, and autonomy. 

In the intervention, among the different activities, software for playing activity, “*Autismo e competenze cognitivo-emotive*”, was utilized, which focuses on potentiating cognitive and emotional competences in children with difficulties in social relations and emotions. It represents a house with different activities for each room: for example, in the kitchen, the subject was asked to recognize a child’s emotion when he received or did not receive the object of desire; in the bedroom, the activity was focused on second-order beliefs; in the attic, the subject was asked to recognize emotions through the eyes.

Half of the sessions were focused on understanding emotions through the use of drawings, pretend play, and short movies chosen to represent happiness, anger, sadness, fear, and disgust. Toward the end of the training, when the children had already mastered these basic discrete emotions, they were presented with more complex ones, such as shame, hope, disappointment, and avarice. The remaining sessions focused on theory of mind, using pretend play and jokes, favoring the understanding of first- and second-order beliefs.

In Table 1, the different types of interventions focused on either emotions or theory of mind are represented. Since the meetings were biweekly, both types of interventions were performed alternately every week. Children’s responses and behaviors were assessed after two months.

## 3. Results

A total of 6 children with a mean (SD) age of 4.22 (0.79) participated in this study. Half of the children were female. Two of the six children had an immigrant background and presented a developmental language disorder (DLD), while others had ASD. Main characteristics for each child are presented in Table 2.

Children attended an average of 14.2 sessions out of 17 in three months of training. After treatment, the total scores of the NEPSY, TEC, and SCQ were compared. NEPSY S02 section showed general improvement that reached statistical significance (Table 3), while, for the other outcomes, we did not find statistical differences. 

## 4. Discussion

The aim of our study was to train preschool children with the assumption that early intervention could improve socialization skills and reduce behavioral problems, thus improving the quality of their relationships and the quality of life [35]. The current study focused on improving conceptual understanding of ToM and emotions in children with ASD or DLD. The reasoning behind this group selection was so that children with both diagnoses could benefit from the interaction with each other; in particular, we believed that DLD children could assist in the process of emotions recognition in ASD children, while the language abilities of the latter could enrich the vocabulary of the former.

Regarding the recognition of emotions, in the early phase of training, children were asked to watch a cartoon representing basic discrete emotions and draw the main characters afterwards. They could represent sad, angry, and happy faces. The adoption of role-playing, psychomotor group activities, and the use of a PC game resulted in a modest improvement in emotion recognition and understanding, although this is not consistently observed in all outcome measures. The efficacy of psychomotricity in emotional behavior is reported in De Lourdes Crò study [36], with positive effects in resilience capacity, dealing with problems, and controlling emotions. In addition, the use of computer-assisted training showed promising results; for instance, Rice LM and colleagues [37] demonstrated that children who received computer-assisted training improved their affect recognition and mentalizing skills, as well as their social skills. Our results showed a statistically significant improvement in the NEPSY S02, which measures emotion recognition ability. In this regard, a recent systematic review [38] confirms that the effects of emotion recognition training in ASD showed significant improvement compared to the control group. Measures designed to capture emotion recognition skills and changes in those skills generally assessed the ability to recognize emotions from facial expression, voices, or postures. These studies used a variety of outcome measures intended to evaluate emotion recognition, including the affect recognition subtest of the NEPSY-II, the Face Task, and photographs from Ekman’s Pictures of Facial Affect [38]. However ToM skills did not improve, and this can be explained by different etiologies. A study conducted on ToM training with children aged 8–13 showed, as main benefits of this intervention, improvements in empathic behavior and social understanding concepts that are explicitly addressed in the training, with little generalization to other aspects of social behavior [7]. Fisher and Happe confirmed that ASD trained children compared to a control group did not show a significant improvement in daily life theory of mind use [50]. Another RCT instead showed the effect of a computer program for training emotion recognition. Results suggested that cognitive abilities improved compared to a control group on emotion recognition in cartoons and second-order ToM reasoning, but not on their recognition of facial emotional expressions [51]. However, it is important to consider that ToM development is strongly affected by non-heritable and social/environmental factors [52,53].

Hence, TOM training could have an important role in the improvement of this ability, but perhaps there should be a more lasting, stable intervention that includes individual sessions and psychoeducative training for parents, teachers, and the whole social environment of the child. It is important to point out that children with ASD, DLD, and attention deficit hyperactivity disorder (ADHD) show very similar emotion recognition skills and are significantly delayed in their development of these skills [39]. In this regard, evidence suggests that ASD, obsessive-compulsive disorder (OCD), and ADHD are classically associated with poor face-processing skills and difficulties in understanding emotions. A recent study, which included these three clinical pediatric groups, compared with typically developing controls, was conducted by using dynamic faces (happy, angry) and dynamic flowers, presented in 18 pseudo-randomized blocks while fMRI data were collected with a 3T MRI. Children with ASD, ADHD, and OCD differentiated less between dynamic faces and dynamic flowers, with most of the effects seen in the occipital and temporal regions, suggesting that emotional difficulties shared in NDDs may be partly attributed to shared atypical visual information processing [54]. Another similar study conducted by using a morphing technique demonstrated a faster response time in neurotypical children compared to ASD and ADHD children, with ADHD participants performing better than ASD participants on the same task [55]. These studies have shown that there is an impairment of emotion recognition in neurodevelopmental disorders, leading to our decision to work on this skill in our neuro-psychomotor training. We choose to use tools such as videos, dynamic faces, drawings, cards, and software because they are easily usable and accessible even for non-verbal children, are easy to provide and comprehend, and are useful in detecting the real abilities of the child in a direct way. Our study provides information about impaired emotion recognition, how to improve this skill in ASD and DLD, and consequently, how to select intervention procedures. At the end of the three-month training, all children could easily understand first-order false beliefs, they could conceptually recognize basic discrete emotions (joy, anger, disgust, sadness, and fear), and, with some guidance, they could recognize more complex ones (shame, hope, disappointment, and greed). The quality of role-play improved in all children, as it was enriched by more details, imaginary objects, and more complex story lines.

Despite the promising results, our study has some limitations. First, the very limited sample size does not allow generalization of findings. Furthermore, different diagnoses can affect the results for ToM skills. However, the explorative nature of the research lays the foundation to set up a more structured study with a larger sample. Furthermore, we did not explore gender differences among children, while heterogeneity of behavior has been documented [18]. In the end, it could be useful to evaluate parents’ and/or teachers’ perspective on children’s behavior. Further studies with a larger population of children should consider these aspects. The present protocol and neuro-psychomotor approach might also be applied in different health conditions, such as children with intellectual disability and ADHD. Further studies should investigate the possibility of applying the present intervention protocol in different neurodevelopmental disorders, involving parents, and promoting a family-centered approach.

## 5. Conclusions

Early neuro-psychomotor therapy is a promising approach for preschool children with neurodevelopmental disorders. The improvement in socialization skills and the reduction of behavioral problems could ameliorate the quality of the relationship and the quality of life.

## Figures and Tables

**Table 1 children-09-01142-t001:** Interventions focused on emotions and theory of mind.

Interventions Focused on Emotions	Interventions Focused on Theory of Mind
Cartoons or short movies focused on basic discrete emotions: anger, joy, disgust, fear, and sadness Discussion about the cartoon watched, e.g., what the character felt and why Associate a color with an emotion: yellow for joy, red for anger, green for disgust, violet for fear, and blue for sadness Mime the emotions seen in the cartoon Music video introducing complex emotions: shame, hope, disappointment, and greed. Drawings: - cartoon’ characters - another member of the group experiencing an emotion - emotion felt - music video character Color a printed thermometer, representing the intensity of an emotion Picture of complex emotions followed by questions Tell when they experienced the emotion represented by a colored hula hoop, following the color code PC game “Autismo e competenze cognitivo-emotive”	Cartoons or short movies focused on false beliefs and understanding of behaviors Discussion on cartoon watched, e.g., what is the character doing and why Enactment of the Sally-Anne test Pretend play with toy kitchen, fake tools or fake food according to child’s ability Role play: - enactment of “Little red riding hood” - doctor, patient, and parent - prince, princess, ghost, witch - False-belief joke using an empty snack package - Games started autonomously by the children - pretend play - Sally-Anne test enactment PC game “Autismo e competenze cognitivo-emotive”

**Table 2 children-09-01142-t002:** Characteristics of participants.

	Child A	Child B	Child C	Child D	Child E	Child F
**Diagnosis**	DLD	ASD	ASD	DLD	ASD	ASD
**Age**	3.11	4.2	3.11	4.10	4.11	4.2
**Ethnicity**	Latino	Italian	Italian	Romanian	Italian	Italian
**Family history**	No family history of ASD or DLD	No family history of ASD or DLD	No family history of ASD or DLD	No family history of ASD or DLD	Family history for ASD and DLD	No family history of ASD or DLD
**Cognitive evaluation**	Griffiths III Development score: 45 months equivalent age;	Griffiths III Development score: 47 months equivalent age;	Griffiths III Development score: 29 months equivalent age;	GMDS-ER Developmental age: 40.5 months equivalent age;	GMDS-ER: Developmental age: 24.5 months equivalent age	WPPSI-III verbal IQ: 100 Performance IQ: 93 Processing speed index: 94

**Table 3 children-09-01142-t003:** Pre-post differences.

Wilcoxon Signed Rank Test	NEPSY S01	NEPSY S02	TEC	SCQ
z	−1.21	−1.99	−0.44	−1.22
Sig (2-tailed)	0.22	0.04 *	0.65	0.22

* *p* < 0.05; NEPSY Neuropsychological Assessment for Child; TEC Test of Emotion Comprehension; SCQ Social Communication Questionnaire.

## Data Availability

Not applicable.

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
