# Peer review of "Early Neuro-Psychomotor Therapy Intervention for Theory of Mind and Emotion Recognition in Neurodevelopmental Disorders: A Pilot Study"

_children, 2022, doi:10.3390/children9081142_

Round 1

Reviewer 1 Report

My suggestions are as follow:

1) The authors use two terms of 'Theory of mind' and 'emotion recognition' in title, however, introduction is lack of review on emotion recognition in paediatric neuro-developmental disorders. So, the authors should mention definition and models of emotions in neuro-scientific point of view (please refer and cite the paper, doi:10.1007/s12021-022-09579-2). After than, the authors should clearly state their motivation on emotion recognition in the present study.

2) In METHODS,

i) A flow-chart or a graphical abstract should be provided in order to make understandable what the researchers did in the proposed study in accordance with emotion recognition.

ii) The following questions have to be answered:

Which emotional states are included in detecting particular neuroscience-developmental disorders ? How did the authors decide skills of the patients in emotion recognition ?

iii) In the same manner, the patient groups should be explained in a table rather than giving long lists.

3) In ABSTRACT,

what does mean the sentence as 'early neuro-psychomotor therapy seems to be a promising 21 approach for emotion recognition in children with neurodevelopmental disorders.' ?

is there any criterion to say this finding?

4) The patients were described into many groups (from A to F). So, ABSTRACT should be modified in order to mention all of them.

5) The authors referred their previous papers in explaining the ethical approval info. Please write both approval date and title of the study approved as well as the name of ethical committee in the relevant sub-section. The papers including the data approved can also be referred with respect to the motivations of them.

6) In DISCUSSION, international terminology should be followed in referring emotional states: 'simple emotions' should be replaced by 'basic discrete emotions'.  The authors should explain the reason in use of three discrete emotions as sadness, angry and happiness. In pediatric psychiatry, the most important emotional states are fear and anger. Sadness and happiness are contrasting emotional states. So, those four emotions are studied in detecting neurodevelopment disorders in literature. The authors should review such papers in DISCUSSION and explain their principles in deciding emotional recognition skills in Children.

Author Response

POINT 1) The authors use two terms of 'Theory of mind' and 'emotion recognition' in title, however, introduction is lack of review on emotion recognition in paediatric neuro-developmental disorders. So, the authors should mention definition and models of emotions in neuro-scientific point of view (please refer and cite the paper, doi:10.1007/s12021-022-09579-2). After than, the authors should clearly state their motivation on emotion recognition in the present study.

RESPONSE 1 Thank you for your observations. We added a moore specific intorduction introducing the reference you suggested and other relevant research papers. Please see introduction section lines 52-72.

POINT 2) In METHODS,

RESPONSE 2

i) A flow-chart or a graphical abstract should be provided in order to make understandable what the researchers did in the proposed study in accordance with emotion recognition.

R. We added a table swith dettailed specific intervention for both ToM and Emotion recognition. 

ii) The following questions have to be answered:

Which emotional states are included in detecting particular neuroscience-developmental disorders ? How did the authors decide skills of the patients in emotion recognition ?

R. Thank you for your consideration

iii) In the same manner, the patient groups should be explained in a table rather than giving long lists.

R. Thank you, we added a table in results section

POINT 3) In ABSTRACT,

what does mean the sentence as 'early neuro-psychomotor therapy seems to be a promising 21 approach for emotion recognition in children with neurodevelopmental disorders.' ? is there any criterion to say this finding?

RESPONSE 3 Agree wit hyou, we modified in: Despite limited sample, early neuro-psychomotor therapy improves emotion recognition skills in children with neurodevelopmental disorders. However, considering the explorative nature of the study, findings should be interpreted with caution. 

POINT 4) The patients were described into many groups (from A to F). So, ABSTRACT should be modified in order to mention all of them.

RESPONSE 4 Considering this is a pilot study, we opted to mantain general consideration in the abstract. We provided specific information in main text

POINT 5) The authors referred their previous papers in explaining the ethical approval info. Please write both approval date and title of the study approved as well as the name of ethical committee in the relevant sub-section. The papers including the data approved can also be referred with respect to the motivations of them.

RESPONSE 5 We added ethical approval in the manuscript please see specific sub-section.  

6) In DISCUSSION, international terminology should be followed in referring emotional states: 'simple emotions' should be replaced by 'basic discrete emotions'.  The authors should explain the reason in use of three discrete emotions as sadness, angry and happiness. In pediatric psychiatry, the most important emotional states are fear and anger. Sadness and happiness are contrasting emotional states. So, those four emotions are studied in detecting neurodevelopment disorders in literature. The authors should review such papers in DISCUSSION and explain their principles in deciding emotional recognition skills in Children.

RESPONSE 6

Thank you. We modified "simple emotions" as you suggested "basic discrete emotions" and we uniformed terms in the manuscript. 

At this regard, evidence suggests that Autism spectrum disorder (ASD)  and others neurodevelopmental disorders are classically associated with poor face processing skills and difficulties in understanding emotions. 

These studies have shown that there is an impairment of emotion recognition in neurodevelopmental disorders, bringing our decision to work on this skill in our neuropsychomotor training. we have chosen to use tools such as videos, dynamic faces, drawings, cards and software because they are easily usable and accessible even for non-verbal children, easy to provide and comprehend, useful to detect the real abilities of the child in a direct way. We specified more in depth in discussion section. 

Reviewer 2 Report

This study examined the effect of early neuro-psychomotor therapy on improving Theory of Mind (ToM) skills and emotion recognition in children with neurodevelopmental disorders.

It is important to enhance ToM skills and emotion recognition in children with neurodevelopmental disorder. However, this study did not contain clear introduction for neuro-psychomotor therapy and its possible mechanisms for its effects on enhancing ToM skills and emotion recognition. It made the study lack of the basis to support. Further introduction for neuro-psychomotor therapy, the mechanisms, and hypotheses of conducting this study should be added in Introduction section.

Moreover, neurodevelopmental disorders contain a heterogeneous group of diseases with various etiologies. This study focused on autism spectrum disorder and language developmental disorder. These two disorders had a series of differences in their etiologies, symptoms, and prognoses. Further introduction is needed to explain the influence of recruiting children with various diagnoses on measuring the effects of intervention.

Emotion recognition but not ToM skills was improved. The possible etiologies should be introduced.

The authors also needed to add introduction for further revisions of intervention programs based on the results of this study.

It is also needed to propose further directions of study to examine the effects of neuro-psychomotor therapy.

Author Response

POINT 1 It is important to enhance ToM skills and emotion recognition in children with neurodevelopmental disorder. However, this study did not contain clear introduction for neuro-psychomotor therapy and its possible mechanisms for its effects on enhancing ToM skills and emotion recognition. It made the study lack of the basis to support. Further introduction for neuro-psychomotor therapy, the mechanisms, and hypotheses of conducting this study should be added in Introduction section. he authors also needed to add introduction for further revisions of intervention programs based on the results of this study.

RESPONSE 1: Thank you for your suggestions. We added description of neuro-psychomotor therapy and added references. Please see page 2 lines 73-83

POINT 2 Moreover, neurodevelopmental disorders contain a heterogeneous group of diseases with various etiologies. This study focused on autism spectrum disorder and language developmental disorder. These two disorders had a series of differences in their etiologies, symptoms, and prognoses. Further introduction is needed to explain the influence of recruiting children with various diagnoses on measuring the effects of intervention.

RESPONSE 2: PWe agreed with you, we specified more in depth reasons for including together children with Development language disorder and ASD. Please see pages 2 and 3, lines 94-102

POINT 3 Emotion recognition but not ToM skills was improved. The possible etiologies should be introduced.

RESPONSE 3. Thank you for your suggestion. We add in discussion and limitation sub-section.

POINT 4 It is also needed to propose further directions of study to examine the effects of neuro-psychomotor therapy.

RESPONSE 4. Agree with you, thank you. We added  further direction for neuro-psychomotor therapy intervention. Please see lines 254-260

Round 2

Reviewer 1 Report

The suggestions have been satisfied in the revised version of the manuscript.

Reviewer 2 Report

The authors have revised their manuscript. I would like to suggest the editors accepting it for publication.